# An Inhalable Theranostic System for Local Tuberculosis Treatment Containing an Isoniazid Loaded Metal Organic Framework Fe-MIL-101-NH2—From Raw MOF to Drug Delivery System

**DOI:** 10.3390/pharmaceutics11120687

**Published:** 2019-12-17

**Authors:** Gabriela Wyszogrodzka-Gaweł, Przemysław Dorożyński, Stefano Giovagnoli, Weronika Strzempek, Edyta Pesta, Władysław P. Węglarz, Barbara Gil, Elżbieta Menaszek, Piotr Kulinowski

**Affiliations:** 1Department of Pharmacobiology, Faculty of Pharmacy, Jagiellonian University Medical College, Medyczna 9, 30-068 Kraków, Poland; gabi.wyszogrodzka@gmail.com (G.W.-G.); elzbieta.menaszek@uj.edu.pl (E.M.); 2Department of Drug Technology and Pharmaceutical Biotechnology, Medical University of Warsaw, Banacha 1, 02-097 Warszawa, Poland; 3Department of Pharmaceutical Sciences, via del Liceo 1, University of Perugia, 06123 Perugia, Italy; stefano.giovagnoli@unipg.it; 4Faculty of Chemistry, Jagiellonian University, Gronostajowa 2, 30-387 Kraków, Poland; weronika.skuza.s@gmail.com (W.S.); gil@chemia.uj.edu.pl (B.G.); 5Department of Pharmaceutical Analysis, Research Network Łukasiewicz—Pharmaceutical Research Institute, Rydygiera 8, 01-793 Warszawa, Poland; e.pesta@ifarm.eu; 6Department of Magnetic Resonance Imaging, Institute of Nuclear Physics, Polish Academy of Sciences, Radzikowskiego 152, 31-342 Kraków, Poland; wladyslaw.weglarz@ifj.edu.pl; 7Institute of Technology, Pedagogical University of Cracow, Podchorążych 2, 30-084 Kraków, Poland; piotr.kulinowski@up.krakow.pl

**Keywords:** theranostic agent, tuberculosis (TB), local pulmonary delivery, isoniazid (INH), metal-organic frameworks (MOF), personalized inhaled therapy, inhaled microparticles, inhalable spray-dried powder blends, ultrashort echo time magnetic resonance imaging (UTE MRI), porous lung tissue phantom

## Abstract

The theranostic approach to local tuberculosis treatment allows drug delivery and imaging of the lungs for a better control and personalization of antibiotic therapy. Metal-organic framework (MOF) Fe-MIL-101-NH2 nanoparticles were loaded with isoniazid. To optimize their functionality a 2^3^ factorial design of spray-drying with poly(lactide-*co*-glycolide) and leucine was employed. Powder aerodynamic properties were assessed using a twin stage impinger based on the dose emitted and the fine particle fraction. Magnetic resonance imaging (MRI) contrast capabilities were tested on porous lung tissue phantom and ex vivo rat lungs. Cell viability and uptake studies were conducted on murine macrophages RAW 246.9. The final product showed good aerodynamic properties, modified drug release, easier uptake by macrophages in relation to raw isoniazid-MOF, and MRI contrast capabilities. Starting from raw MOF, a fully functional inhalable theranostic system with a potential application in personalized tuberculosis pulmonary therapy was developed.

## 1. Introduction

The deceptive nature of tuberculosis (TB) contributed to its long-time reputation of curable and defeated disease. Alongside poor health care settings and treatment control, limited understanding of TB infection peculiarities and the complex host-pathogen crosstalk have crippled progresses in the field. Current therapies rely on cumbersome, lengthy multidrug treatments that, albeit still effective at early infection stages, lose efficacy rapidly at later disease phases and on latent infections. A dramatic but inevitable consequence is the alarming worldwide growth of drug resistance that has undermined the sustainability and efficacy of current TB treatments. This scenario also partially explains the general crisis of the antibiotic drug market. On the other hand, a renewed awareness and increased knowledge of TB infection features have encouraged recent efforts towards the development of host-directed approaches and inhaled TB therapies. Undoubtedly, pulmonary anti-TB drug delivery is the logical strategy for maintaining local therapeutically effective concentrations while avoiding massive systemic exposure and consequently reducing side effects [1,2].

However, the efficacy of inhaled antibiotic treatments is strictly dependent on drug deposition patterns within the lungs, which in turn are determined by the formulation performances and pathophysiological factors [3,4]. Even when proper susceptibility screening and optimal formulations are being adopted, disease-related impairment of conductive airways can cause insufficient delivery to the infected sites, resulting in sub-effective local antibiotic levels. Such an issue is detrimental to treatment efficacy, raising serious resistance development concerns. Moreover, the often neglected large individual variability in lung capacity and physiology reduces further the chances of success of inhaled antibiotic therapies [5]. The disease-induced lack of access to poorly-aerated lung regions is hard to detect clinically. In fact, healthy conductive airways may compensate such an impairment, masking the partial improper ventilation to standard functional lung tests, such as spirometry [6].

Therefore, the assessment of infection-related lung function alterations is a priority in order to establish proper countermeasures enabling effective drug deposition within the lungs [5]. To discriminate between the effect of the delivery mechanism and the pharmacological action, it is crucial to (i) properly assess the effectiveness of the treatment, (ii) adjust the appropriate dose, and (iii) prevent false negative treatment results. In this regard, the development of functional lung imaging methods, such as magnetic resonance imaging (MRI), may help to discriminate between drug and pathophysiological effects [7,8].

When diagnosing pulmonary diseases, the MRI can be used to detect disease-induced changes in the lungs [9,10,11]. MRI is especially attractive when dealing with pediatric patients [12] and pregnant women [13]. MRI methods designed for signal acquisition from environments exhibiting very short T_2_/T_2_* relaxation times (property of lung tissue/environment), like ultra short echo time (UTE) and zero echo time (ZTE) [14], have demonstrated their correlations with X-ray computed tomography (CT) [12]. Moreover, UTE has been applied to study aerosol deposition using iron oxide particles ex vivo [15,16] and gadolinium (Gd)-DOTA particles in vivo [17,18].

These techniques may enable treatment personalization, and thus, maximizing therapeutic outcomes by granting control over lung deposition and local drug levels [5,19]. Although theoretically feasible, inhalation treatment personalization through imaging techniques is not an easy task and requires the development of proper tools to record a patient’s conditions and inhalation behavior. In this regard, the concept of theranostic is in the spotlight in personalized medicine [20,21,22].

The term “theranostic” was coined by merging the words therapy and diagnostics. A classical theranostic agent is defined as a system enabling simultaneous monitoring of drug delivery and treatment outcomes so as to allow a proper therapy–patient match [23,24]. Nowadays, nanotechnology has allowed the combination of such features even into a single particle, coupling an effective delivery approach with imaging capabilities [25]. In this regard, functional imaging has the potential to provide new insights into the behavior of inhaled particles by allowing monitoring of deposition patterns upon administration. The possibility of tracking particles after inhalation may allow one to assess the actual drug targeting to the infected sites and personalized dose adjustments. This may grant patient-centered therapeutic regimens and administration strategies.

Metal-organic frameworks (MOFs) are porous materials whose structures are built by inorganic single ion or ion cluster nodes joined together by organic linkers [26,27]. Their characteristic feature is the presence of a metal center, which may be a paramagnetic cation (e.g., iron), responsible for the MRI contrast property [28,29]. In our previous study, we presented the possibility of application of Fe-MIL-101-NH_2_ MOF, consisting of Fe (III) and 2-aminoterephthalic acid, as a controlled release carrier for isoniazid (INH) and an MRI contrast agent [29]. The results suggest that Fe-MIL-101-NH_2_ loaded with the drug can serve as an effective MRI contrast agent and can be used as a foundation for constructing actual drug delivery systems. The in vitro cytotoxicity studies of Fe-MIL-101-NH_2_ confirmed its safety and the accumulation observed in the cell cytoplasm supports a potential application in pulmonary tuberculosis therapy [29].

Despite extensive studies concerning potential pharmaceutical MOF applications [28,29,30,31,32], there is still an unmet need for effective MOF-based drug delivery systems. 

Particularly for inhaled TB therapy, MOFs show a number of limitations that require tailoring of their physico-chemical and solid-state properties. In this regard, our preliminary studies have shown that INH-loaded MOF (INH-MOF) display insufficient aerodynamic performances to stand alone as an inhaled dosage form. Moreover, proper formulation approaches are needed to ensure INH-MOF macrophage uptake, which is a recognized strategy to enable successful targeting of primary TB infection sites and prevention of further infection spreading [33,34].

Therefore, the main goal of this study was to formulate an INH-MOF based theranostic system showing optimal aerodynamic properties using the spray drying technology. For this purpose, hydrophobic and hydrophilic microparticles (MPs) loaded with INH-MOF were prepared to provide appropriate formulation flowability and prolonged drug release. Particles were blended and characterized for their aerodynamic behavior, morphology, in vitro drug release, biocompatibility and MRI contrast capabilities.

## 2. Experimental Section

### 2.1. Materials

Isoniazid, D-Leucine (LC), poly(lactide-*co*-glycolide) (PLGA, Resomer^®^ RG504H), tetrabutylammonium hydroxide solution (tBAH), HPLC-grade acetonitrile (ACN), and trifluoroacetic acid (TFA) were from Sigma-Aldrich (Taufkirchen, Germany). PrestoBlue^TM^ test, a dichloro-dihydro-fluorescein diacetate (H_2_DCFDA) assay kit were purchased from ThermoFisher and ToxiLight from Lonza. Murine macrophages—RAW 246.9 cell line—Were obtained from ATTC. Macrophage Detachment Solution was from Promocell.

### 2.2. MOF Synthesis and Drug Incorporation

Fe-MIL-101-NH_2_ was synthesized according to the procedure reported by Bauer et al. [35]. After drying, MOF was activated under vacuum at 100 °C to remove residual solvent molecules. INH was incorporated into the activated MOF by mixing 5 mL of saturated INH solution in DMF with 1000 mg of Fe-MIL-101-NH_2_ for 12 h. The product was separated by centrifugation.

### 2.3. Preparation of Spray-Dried Particles

LC and PLGA MPs were prepared by spray-drying using the B290 Mini spray-dryer (Buchi, Flawil, Switzerland). For PLGA MPs, MOF (with and without drug) was suspended in a 12 mL of ACN PLGA solution. The total amount of solids was 400 mg and the process parameters were as follows: air flow rate 357 L/h, inlet temperature 75 °C, aspirator rate 20 m^3^/h, feed rate 2.4 mL/min. For LC MPs, LC and MOF were solubilized in 15 mL of water (total amount of solids 300 mg), and spray-drying was performed at the following conditions: air flow rate 473 L/h, inlet temperature 140 °C, aspirator rate 20 m^3^/h, feed rate 2.4 mL/min.

### 2.4. Design of Experiment (DOE)

Blends were prepared by placing appropriate amounts of each powder in a tube (total amount 200 mg) and mixing them for a specified time at a speed of 60 rpm in a coclea mixer kept at an angle of 45°. The effects of MOF loading (factor A), PLGA/LC blend ratio (factor B), and blending time (factor C) were investigated on emitted dose (%ED) and fine particle fraction (%FPF) employing a 2^3^ factorial design (Table 1).

Design Expert version 8.0.1 software (Stat-Ease, Minneapolis, MN, USA) was used to evaluate the effect of the selected factors on the properties of the blend. The selected variables were interpolated with a quadratic polynomial equation (Equation (1)).
(1)$$y=b_{0}+\sum_{i=1}^{k}b_{i}x_{i}+\sum_{i=1}^{k}\sum_{j=1}^{k}b_{ij}x_{i}x_{j}+\sum_{i=1}^{k}b_{ii}x_{i}^{2}+e,$$
where *y* is the response; *b_0_*, *b_i_*, *b_ii_*, and *b_ij_* are the regression coefficients; *x_i_* and *x_j_* are the level of the *i*-th and *j*-th factors; and *e* is the residual random error of the model.

The models were statistically evaluated by ANOVA and response surfaces were built to evaluate the effects of the three select factors on the aerodynamic behavior of the blend.

To define the conditions granting the desired characteristics, the obtained response surfaces were combined by applying the total desirability approach [36], which is represented by Equation (2):(2)f(d(g))=∑i=1Mwidi∑i=1Mwi,
where *f*(*d*(*g*)) is the total desirability, of the *i*-th response, *M* is the number of responses and *w* is the weight of each response specified by the user. Desirability assumes values between 0 and 1 and the maximum desirability corresponds to the conditions that provide compliance with the established quality criteria.

### 2.5. Particle Characterization

#### 2.5.1. In Vitro Aerodynamic Characterization

Aerodynamic properties were assessed according to the European Pharmacopeia Ed. IX, using a twin-stage impinger (TSI). The two stages of the TSI were filled with 7 mL (stage 1) and 30 mL (stage 2) of 50:50 ACN: 0.1 M NaOH solution (extraction medium for placebo MPs) and 50:50 ACN: 0.01 M phosphate buffer, pH 7.4 (for MPs containing INH). In total, 20 mg of powder was loaded into an HPMC capsule (Type 3, Quali-V, Qualicaps^®^ S.A.U, Romania, Spain). The air flow rate was set at 60 ± 5 L/min and emission was performed for 5 s through an RS01 model dry powder inhaler. The %ED and %FPF were calculated using Equations (3) and (4):%ED = emitted dose 100/Nominal dose(3)
% FPF = dose (stage 2) 100/Emitted dose(4)

#### 2.5.2. MOF and Drug Quantification

MOF was analyzed by UV–vis spectrometry using an Agilent 8453 spectrophotometer. Samples were sonicated during 20 s and analyzed (λ_max_ = 325 nm). Calibration was performed in the 25–200 µg/mL concentration range in a 0.1 M NaOH solution (R^2^ = 0.9987).

INH was assayed by HPLC using a HP1050 chromatograph equipped with UV–Vis detector set at 254 nm and a GROM-SIL 120 Diol 250 × 4.6 mm (Grace Davison Discovery Sciences) column. Operating conditions were: column temperature 25 °C; flow rate 1 mL/min; the mobile phase was 70:30 ACN: 0.4 mM tBAH, pH 6. Calibration was performed in the 0.5–16 µg/mL concentration range in water (R^2^ = 0.9997).

#### 2.5.3. Sampling for Content Uniformity and Content Average

In total, 200 mg of formulation powder was spread over the surface of a circle with a 5.5 cm diameter. The circle was divided into three slices and three samples (about 3 mg) were gently taken from each slice and assayed as reported above. The average MOF/INH content was calculated on 9 replicates. Acceptance criteria were percent relative standard deviation (%RSD) ≤5%, and mean ± 10% from target value according to USP 34.

#### 2.5.4. Particle Size and Morphology

Particle size analysis of PLGA MPs was carried out by an Accusizer C770 particle counter (PSS, Santa Barbara, CA, USA) equipped with an autodilution system. Due to LC solubility in water and even partial solubility in organic solvents, the particle size of LC MPs was determined by scanning electron microscopic (SEM) images, using Image J software. An average of 200 particles from at least six different SEM images were counted and size distributions and mean diameters were calculated.

SEM analyses were performed by using a FEG LEO 1525 microscope (LEO Electron Microscopy Inc., Thornwood, NY, USA). The acceleration potential voltage was 1 keV. Samples were placed onto carbon tape coated aluminum stubs and sputter coated with chromium for 20 s at 20 mA (Quorum Technologies, East Essex, UK). Additional observations were conducted on dispersed particles collected from the impinger immediately after actuation. Energy dispersive X-rays spectroscopy (EDX) was performed by detecting Fe and N signals at 10 kV with acquisition times of 20 min. The coating was performed with graphite at 20 mA for 22 s.

### 2.6. In Vitro Drug Release Study

In vitro drug release experiments were performed by the dialysis bag method (12–14 kDa MWCO). A 0.1 M phosphate buffer (pH 7.4) solution was employed as a release medium, using a 50 mL conical Falcon tube incubated at 37 °C. An experiment was conducted for the optimized INH-MOF-loaded PLGA/LC MP blend, PLGA MPs, and LC MPs. Drug release quantification was performed in triplicate by HPLC using the method described above.

### 2.7. Statistical Analysis

The results for statistical analysis are presented as the mean ± %RSD. One-way ANOVA and descriptive statistics (software SigmaPlot, version: 14.0, San Jose, CA, USA) were employed to analyze content uniformity in PLGA/LC MP blends. Dunnet’s and Tukey’s post-hoc methods were employed for the evaluation of differences within and between groups. *p* values of < 0.05 were considered significant.

For the in vitro tests on cells, the bar-graph data show the means and standard deviations of one representative of 6 independent measurements. Significant effects (*p* < 0.05) were determined using the independent *t*-test, also called the two sample *t*-test, an inferential statistical test that determines whether there is a statistically significant difference between the means in two unrelated groups. The *t*-test was used to check differences between the samples cultured in the presence of MOF at different concentrations and control but not to compare different MOF concentrations with each other.

### 2.8. Magnetic Resonance Imaging (MRI)

#### 2.8.1. Preparation of Sponge Phantoms of Lung Tissue

A sponge phantom was prepared to reflect the structure of lung tissue. It consisted of three 2 × 2 × 1 cm^3^ cellulose sponge bricks. Each sponge brick was wet with 1.5 mL of raw MOF water suspension, hermitized, and retained for equilibration. The amount of MOF suspension was chosen to hydrate the sample and left the largest pores liquid free. Sponges containing pure water and the highest MOF concentration were fixed and between them the central one was replaceable. The concentrations of MOF suspensions were 0 (pure water), 0.03, 0.0625, 0.125, 0.25, 0.5, and 0.75 mg/mL. Concentrations of MOF in suspension corresponded to the mass of MOF per sponge volume of 0, 11.25, 23.25, 46.875, 93.75, 187.5, 281.25, and 375 μg/cm^3^. Fixed sponges were used for reproducibility testing, as they allowed us to calculate mean values for six independent measurements (*n* = 6).

#### 2.8.2. Preparation of Lung Ex Vivo Samples

The experiment was performed on excised Wistar rat lungs obtained from animal facility of Faculty of Pharmacy Jagiellonian University Medical College. Appropriate amounts of the optimized INH-MOF-loaded PLGA/LC MP blends, 3.75 and 18.77 mg (equivalent to 10% and 50% of the INH oral dose) were administered into the lung via trachea. The lungs were stored in 0.9% NaCl solution and transferred to 20 mm ID tubes. MOF concentration in the lung was calculated assuming the rat lung volume of 3 mL [37].

#### 2.8.3. MR Imaging and Image Analysis

Magnetic resonance imaging was performed using 9.4 T MRI research system (Bruker Biospin, Rheinstetten, Germany). A 3D ultra short echo time (UTE3D) imaging sequence was used with following parameters: echo time (TE) = 110 µs, repetition time (TR) = 2.345 ms, flip angle = 5°, number of accumulations (NA) = 1, number of radial projections = 52,402, and reconstructed 3D volume image of 160 × 160 × 160 voxels. For sponge phantom imaging, field of view (FOV) was 45 × 45 × 45 mm^3^, and for lung imaging it was 35 × 35 × 35 mm^3^.

3D volume images were imported to Fiji distribution of ImageJ version 1.44 (National Institutes of Health, http://rsb.info.nih.gov/ij/) [38].

Concerning sponge phantoms, each sponge volume was extracted by 3D cropping, and for the cropped volume images the histograms were calculated. The histograms were normalized to the number of voxels in the image volume. The maximum value of each histogram was assessed. The maximum histogram values obtained for MOF wetted samples were expressed as relative to the maximum histogram value obtained for water wetted volume. The results for all measured MOF concentrations were fitted (linear fit) using Microsoft Excel 2010 (Microsoft, Redmond, WA, USA).

3D binary mask of the lung volume was obtained as a result of image segmentation of the original 3D image volume. The supervised image segmentation was performed using a Trainable Weka Segmentation 3D (TWS 3D) plug-in module (https://imagej.net/Trainable_Weka_Segmentation), part of the public domain software, Fiji distribution (https://fiji.sc/) of ImageJ—built on Waikato Environment for Knowledge Analysis (WEKA) data mining and machine learning toolkit [39]. The binary mask and the original image were multiplied (voxel by voxel multiplication). For the segmented volume images, the histograms were calculated and normalized to the total number of counts. Finally, rendered fine structure of the lungs was visualized using 3D Viewer plugin.

### 2.9. In Vitro Study—Cell Viability, Cytoxicity, and Cellular Uptake

RAW 246.9 was grown in Dulbecco’s modified Eagle’s medium, DMEM, with the addition of 10% (*v*/*v*) fetal bovine serum. Cells were cultured in a humidified atmosphere at 37 °C with 5% CO_2_, incubated until an 80% confluent cell monolayer was developed, and detached from the culture flasks using Macrophage Detachment Solution. The experiment wasdivided into two stages. First, cells were plated at the density of 5 × 10^3^ cells/well in a 96-well plate, and after 24 h were incubated with various MOF concentrations (500, 100, 50, 25, and 10 μg/mL). Based on results obtained for pure MOF, 100 μg/mL was chosen as the optimal MOF concentration for further study. In the next step, cells were plated at the density of 200 × 10^3^ cells/well in a 24-well plate. After 24 h of incubation, cells were exposed to blend and a corresponding quantity of INH-MOF at concentrations 500 and 150 μg/mL, respectively.

Cells were incubated for 24 h in standard conditions, and after that, the viability, cytotoxicity, and level of ROS were determined.

Cell viability was examined by resazurin-based reagent PrestoBlue^TM^. Material cytotoxicity was determined by ToxiLight^TM^ BioAssay Kit and ToxiLight^TM^ 100% Lysis Reagent Set.

The effect of materials on the level of reactive oxygen species (ROS) was investigated using dichloro-dihydro-fluorescein diacetate (H_2_DCFDA) assay. As a positive control, 100 μM H_2_O_2_ was used.

The cell viability, toxicity, and ROS generation level were evaluated following the manufacturer’s protocols for fluorescence or luminescence measurement using a microplate reader POLARstar Omega (BMG Labtech, Ortenberg, Germany). Cell morphology was observed under contrast phase inverted microscope CKX53 (Olympus, Tokyo, Japan).

The amount of phagocytosed INH by macrophages after treatment with INH-MOF and the optimized INH-MOF-loaded PLGA/LC MP blend was examined after 0.5, 3, 6, and 24 h. Supernatants were collected, and cells were washed by PBS and lysed by a 5% solution of Triton X-100 for 30 min. To analyze lysates, the gradient reverse-phase high-performance liquid chromatography (RP-HPLC) method was developed. The chromatographic separation of INH from formulation ingredients was carried out on a Kinetex XB C18, 2.6 µm (150 × 4.6 mm) column with a mobile phase composed of A: 0.1% TFA in water and a mobile phase B: 0.1% TFA in ACN. The flow rate was 0.4 mL/min and detection was monitored at 254 nm.

## 3. Results

### 3.1. Experimental Design

The INH content in the INH-MOF used to prepare the formulation was about 12% [29]. The development and optimization of an INH-MOF inhaled theranostic formulation was carried out by applying a 2^3^ factorial design. In order to meet the fundamental requirements of proper nebulization, deposition, and pulmonary retention, INH-MOF was encapsulated in hydrophobic poly(lactide-*co*-glycolide) (PLGA) MPs by spray-drying and blended with spray-dried INH-MOF-loaded D-leucine (LC) MPs.

The aerodynamic properties of the blends were investigated in terms of emitted dose (%ED; the amount of powder released from the device) and fine particle fraction (%FPF; the amount of powder deposited in the 2nd stage of the TSI that corresponds to the lower lung airways). The design space for formulation optimization is presented in Table 1. The independent variables consistED of MOF content, blending ratio, and blending time. The responses were %ED and %FPF, which describe the aerodynamic behavior of the PLGA/LC MP blends.

As pointed out by the coefficient estimates calculated for the two quadratic models (Appendix A), the percentage of LC MPs in the blend had a dominant effect on %FPF. In fact, %FPF reached values >50% when the PLGA/LC ratio was <30% (Figure 1). Blending PLGA MPs with LC MPs improved, considerably, the respirability of the powders compared with the poor behavior of MOF and the PLGA MPs alone (%FPF < 30%), confirming the fundamental glidant role of LC, as also observed by others [40].

The concomitant loading of INH-MOF in both the hydrophobic and hydrophilic MPs was thought to limit the drug dilution effect resulting from the use of multiple excipients.

An important and non-linear effect of MOF loading on %FPF was observed. In contrast, %ED was not much influenced by MOF loading with values always higher than 91%, regardless of the amount of MOF encapsulated. The best aerodynamic performances were achieved at the MOF content value of about 30% *w*/*w*. Blending time was generally uninfluential, on both %FPF and %ED (Appendix A). Overall, the models obtained indicated that target conditions could be fulfilled at about 30% MOF content and a PLGA/LC ratio between 30% and 40%. These working coordinates provided blends with a theoretical %FPF close to 60% and an %ED >93%.

Therefore, the final blending conditions were established at 30% MOF-INH loaded MP blended, at 30% PLGA/LC ratio for 15 min. The measured values of %FPF = 56.7 (%RSD = 2.30) and %ED = 90.9 (%RSD = 1.18) indicated good aerodynamic powder properties and confirmed model predictivity.

### 3.2. Powder Characterization

#### 3.2.1. Morphology and Size

MOF loading in LC and PLGA MPs produced rather irregular particles due to the presence of bipyramidal MOF crystals (Figure 2).

The encapsulation of MOF crystals produced fragile MPs, particularly LC MPs, that were partially fragmented during blending (Figure 3a). This resulted in broad particle size distributions with average volume-weighted diameters around 11 μm for MOF and 20 µm for PLGA MPs, while number-weighed diameters were from 0.9 µm for MOF to 4.2 µm for PLGA MPs and 6.7 µm for LC MPs (Figure 3b). Such size features suggest a compliance with pulmonary delivery requirements. In fact, it is suggested that the small particle size below 5 µm may grant better inhalation efficacy than larger particle size aerosols, owing to an enhanced penetration and retention in the lungs even in the presence of airway narrowing [41,42]. In particle size distribution of INH-MOF PLGA MPs (Figure 3b) two populations were observed: one corresponding to the particles and another to the free INH-MOF, which indicates the presence of MOF also on the surface of the INH-MOF PLGA MPs.

#### 3.2.2. In Vitro Aerodynamic Characterization and Content Uniformity

The TSI study showed that the MPs blend aerodynamic behavior was a measure of the presence of large clusters in the powder. In fact, particle clusters >10 µm were deposited in the stage 1 of the TSI, while the smaller aggregates (below 10 µm) were all found in the stage 2 (Figure 4). Therefore, this size cutoff was considered discriminative in determining respirability of the obtained blends.

As reported in Table 1, a good content uniformity was measured either in the blends or the PLGA and LC MPs (Appendix A), with only the samples from runs 3, 5, and 6 showing %RSDs >5. However, only the blends from runs 9–11 had the target value within ± 10% of the calculated mean (Appendix A). In all the other cases, the actual content was significantly higher than the nominal one, especially at 40% theoretical loading. The measured homogeneity was consistent with the homogeneous distribution recorded by EDX mapping (Figure 5). In fact, INH-MOF was partially evenly distributed on the surface of particles (Figure 5), which confirms observation in the particle size distribution of INH-MOF PLGA MPs (Figure 3b). High MOF content and an even distribution are important to ensure tracking capacity of the inhaled blend. The average content of INH in the final blend was 3.7% *w*/*w* (%RSD = 3.7).

### 3.3. In Vitro Drug Release Study

The drug was released faster during the initial 10 h than in the subsequent time period (Figure 6). As a result of the excipient properties, a higher INH release was observed for LC MPs compared to PLGA MPs (80% versus 59%, after 24 h). Since LC is water-soluble, a relatively fast LC MP dissolution was expected with simultaneous drug release. Therefore, the release from LC MPs depended on the intrinsic affinity of INH to MOF and MOF matrix degradation [28]. In fact, although LC MPs dissolve immediately, INH dissolution was controlled by MOF, especially at neutral pH, which determined the slow INH release. On the other hand, the hydrophobic nature of PLGA moderately reduced the initial phase of drug dissolution. Drug release from PLGA MPs was assumed to be the result of more complex hydration and diffusion processes. The hydrophobic properties of PLGA may explain the incomplete release of INH from PLGA MPs [43].

Blending of hydrophilic and hydrophobic particles is to provide dual release of a drug. LC is very soluble in contact with fluids and provides quicker release of INH-MOF compared to PLGA MPs. The expected influence of both PLGA and LC MP properties on the INH release profile from the blend produced a biphasic behavior with about 34% and 67% of drug release after 1 and 24 h, respectively. The release was complete after about 4 days either for LC MP or the optimized blend.

### 3.4. MRI Contrast Capabilities

For all samples, the maximum histogram value obtained for an exchangeable sponge was between the maximum of the histogram obtained for the water wetted sponge and the maximum obtained for the sponge containing 375 µg of MOF per cm^3^. An example of the resultant histograms obtained using exchangeable sponge phantom containing 11.25 µg of MOF per cm^3^ is presented in the left panel of Figure 7d.

In the right panel of Figure 7d final results are presented; i.e., the dependence of relative maximum histogram value on MOF concentration in the sponge, as expressed in dry MOF mass per volume of the sponge brick (where the value 1 corresponds to the maximum histogram value for water wetted sponge brick). At the lowest MOF concentrations, i.e., between 11.25 and 93.75 µg/cm^3^, the difference between the maximum histogram values for pure water and the MOF wetted sponges was about 10–30% of the water histogram’s maximum. For the highest measured MOF concentration (375 µg/cm^3^) the maximum of the histogram was as low as 0.6 of water maximum histogram value.

Concerning reproducibility, for water wetted sponge brick the mean maximum histogram value was 17.026 × 10^−3^ ± 0.937 × 10^−3^. For the 375 µg/cm^3^ bricks the average histogram maximum value was 10.254 × 10^−3^ ± 0.824 × 10^−3^ in absolute values and 0.575 ± 0.046 relatively to the water histogram maximum.

Figure 8 shows the MRI results obtained for extracted rat lungs treated with optimized INH-MOF-loaded PLGA/LC MP’s blend. An example of one slice of the 3D image volume for untreated rat tissue is presented in the left panel of Figure 8a. A rendered 3D lung image based on segmented volume is presented in the right panel of Figure 8a. Histograms of the three image volumes, i.e., untreated lungs, treated with formulation dose fixed at 10% of an oral dose (MOF concentration in the lung of 375 μg/cm^3^) and treated with dose fixed at 50% of oral dose (MOF concentration in the lung of 1875 μg/cm^3^), are presented in Figure 8b. Histograms of the MR images of rat lungs followed the same pattern as obtained on sponge fantom; i.e., decrease in max histogram value and overall expansion towards higher image intensities with an increase in MOF concentration per volume of the sample. Moreover, rendered fine details of lung structure are shown in Figure 8c. Upper left panel of Figure 8c presents structural details for untreated lungs and upper right panel of Figure 8c for the rat treated with dose containing MOF at a concentration of 375 μg/cm^3^. The lower panel of Figure 8c presents fine lung structural details for rat treated with dose containing MOF at a concentration of 1875 μg/cm^3^. The comparison of the untreated lungs and the lungs treated with final formulation at various MOF concentrations showed enhancement of the bronchi structure as a result of the MOF presence in the formulation powder at the surface of various parts of the tract (e.g., at bronchi walls). Not only primary and secondary, but also tertiary bronchi were able to visualize (inside the yellow ring in Figure 8c).

### 3.5. Time-Dependent Cellular Uptake

Our previous research proved that MOF MPs could be phagocytosed by fibroblasts [29]. On this basis, in this work optimized INH-MOF-loaded PLGA/LC MP blend’s uptake in macrophages was investigated to assess the potential to target the primary site of TB infection and residency. Time-dependent cellular uptake was investigated in RAW cells treated by the optimized, INH-MOF-loaded PLGA/LC MP blend and INH-MOF for 0.5, 3, 6, and 24 h. The amount of INH encapsulated in both samples was about 20 µg per well. Both INH-MOF and the optimized INH-MOF-loaded PLGA/LC MP blend were internalized by macrophages and subsequently released inside the cells. An increase in INH concentration was observed at each measurement time point, reaching after 24 h the maximum value of 16.66 ± 0.01 pg/cell for blend and 13.20 ± 0.01 pg/cell for INH-MOF. The percentage of accumulated INH (after 24 h) in reference to total applied dose was 21.13% ± 0.02% for the optimized INH-MOF-loaded PLGA/LC MP blend and it was 16.16% ± 0.36% for INH-MOF (Figure 9).

### 3.6. Cytotoxicity Study

In the presence of MOF in low to moderate concentrations (10–50 μg/mL), no effect on cell viability was observed (Figure 10a). For the lowest concentrations (10 and 25 μg/mL) an increase in metabolic activity after 24 h of incubation was observed when compared to the control (116% ± 2.5% and 113% ± 1.6%, respectively). After prolongation of the incubation time, no further increase is observed. At the highest concentration (500 μg/mL), the presence of MOF led to decrease of the cell viability by 52% (±2.2%) or 51% (±1.5%), respectively, after 24 or 72 h, in comparison with the control group. However, after 72 h the number of cells was still higher than that observed after 24 h; therefore, in MOF’s presence the proliferation and activity of macrophages were slowed down but not inhibited. On this basis, it may be determined that the highest concentration of MOF which can be considered as “safe” for the cells is 100 µg/mL. The difference between the viability of cells incubated with MOF and the control group was only 5.1% (±2.3%) after 24 h and 8.9% (±1.2%) after 72 h. The cytotoxicity of pure MOF was increased by 12.9% after 24 h and by 22.8% after 72 h, as compared to the untreated cells.

Subsequently, the influence of INH-MOF and the optimized INH-MOF-loaded PLGA/LC MP blend on the viability of the macrophages (Figure 11a) was determined. The powder concentrations were calculated on MOF content determined as safe for the cells (ca. 100 µg/mL). After 24 h, an upward trend in cell viability, measured as the mitochondrial activity of macrophages, was observed for both INH-MOF and the optimized INH-MOF-loaded PLGA/LC MP blend. This decrease was not substantial enough to confirm the cytotoxic effect of MOF on macrophages. Additionally, there was observed doubled level of produced reactive oxygen species (ROS) for cells incubated with INH-MOF and the optimized INH-MOF-loaded PLGA/LC MP blend compared to untreated cells. However, it was still much lower (13 times) than ROS concentration obtained for the positive control group (treated with 100 μM H_2_O_2_ solution) (Figure 11a). The morphology of macrophages changed—After 24 h, cells become more rounded and some of them were swollen and a phagocytized MPs were observed inside (Figure 11b).

## 4. Discussion

As reported previously, merely aerosolizing anti-TB drugs may not be sufficient to provide effective therapy [44]. For reaching deep lung levels and efficient antibacterial action, drugs need to be formulated into delivery systems ensuring their uptake into alveolar macrophages where bacilli are located. The obtained optimized INH-MOF-loaded PLGA/LC MP blend showed good aerodynamic properties (%FPF = 56.7) and the ability to penetrate into the interior of macrophages (INH uptake after 24 h was 21% of the dose).

The MOF’s safety was confirmed in our previous and current in vitro studies, which showed that MOF did not cause a cytotoxic effect and did not inhibit L929 fibroblast proliferation even at a high concentration (1.25 mg/mL) [29]. Moreover, iron carboxylate MOF’s safety has been proven in in vivo studies conducted by Horcajada et al. [28], with complete MOF elimination and no side effects (after 3 months from i.v. injection, 220 mg/kg). Additionally, we proved that MOF, INH-MOF, and the optimized INH-MOF-loaded PLGA/LC MP blend did not cause a cytotoxic effect on macrophages. In the previous study, MOF incubated for 24 h with L929 fibroblasts was phagocytosed and clearly visible inside the cells in microscopic pictures, which suggests that MOF could release the drug inside the cells; however, there is no evidence of INH uptake by macrophages [29]. In this study, we quantified the amount of INH inside the macrophages and examined the formulation effect on the cellular uptake. Results (Figure 8) indicate that both INH-MOF and the optimized INH-MOF-loaded PLGA/LC MP blend were internalized by the macrophages and higher uptake was observed for the blend (16.66 ± 0.01 pg/cell for blend and 13.20 ± 0.01 pg/cell for INH-MOF after 24 h). An increase of INH concentrate on inside the macrophages was observed during the whole experiment, which is not surprising, since PLGA activates alveolar macrophage phagocytosis and is often used to increase the local concentration of drugs within cells [45,46,47].

The observed increase in the ROS level after 24 h of incubation with the blend, in concentrations as high as 500 µg/mL, may support the antibacterial activity of INH against *Mycobacteria*. Additional effects of the Fe^3+^ ions against the bacterial growth and survival have been already reported [48,49,50,51,52].

The observed release of INH is a result of the concerted action of PLGA and INH-MOF and LC and INH-MOF. There are a few levels in drug dissolution: (1) INH release from INH-MOF located on the PLGA MPs surface (Figure 3b); (2) INH-MOF release from microparticles (LC or PLGA); and (3) INH release from MOF. Release of INH from LC MPs roughly resembled release of INH from pure INH-MOF NP’s (see Figure 6 and Figure 4 in Wyszogrodzka et al.) [31]. Naturally, the choice of PLGA as a long-term releasing polymer caused further retardation in INH release from INH-MOF NPs and increased drug residence time in the lungs. The choice was governed by the rather limited arsenal of accepted materials for lung drug delivery. To date, this is perhaps the major limitation to commercialization of new and more effective inhaled formulations, able to grant sufficient local and systemic antibiotic levels [34].

The selection of excipients for inhaled formulations requires serious consideration of the therapy requirements. In fact, the use of excipients in pulmonary drug delivery, albeit necessary to provide the required aerodynamic and rheological properties, can be a considerable pitfall for inhaled antibiotic therapies, since it can determine the need for an excessive aerosolized mass of material to reach the required effective dose. Whether the goal is targeting pathogen’s niches or grant local and systemic therapeutic levels, a high drug concentration and low amount of excipient are desirable, because the quantity of material that the patient is able to inhale without adverse events sets the size of the therapeutic window [53]. Naturally, if the aim of the therapy is to target macrophages, the excipients should provide a slower dissolution rate, for prolonged internalization by phagocytosis [33]. Unfortunately, although there are many examples of inhaled INH formulations in the literature, there is still no inhaled anti-TB product available on the market, and therefore dose-determining criteria are lacking [44,54]. In light of such considerations, our approach was based on the blending of two kinds of particles, each one embedding the drug-MOF complex, in order to fulfill the above-mentioned requirements for pulmonary drug delivery and limit the drug dilution effect which undermines the use of microencapsulated drugs in inhaled therapies. In our case, the drug delivery and MRI tracking agent double role of the optimized MP blend may contribute to ensuring the required delivered dose and helping to determine the clinical relevance of the proposed approach owing to the possibility of MRI tracking. This fundamental property was evaluated by using sponges as a heterogenous lung phantom.

Such lung simulating systems, albeit clearly rather simplified, have been applied in several studies because of the pore size distribution—Ranging from sub-millimeter to a few millimeters with sub-millimeter pores located in the walls of large pores [55]. They have been used as phantoms for lung ultrasound surface wave elastography [55] and for lung MR imaging [56]. Different MRI pulse sequences, including FSE (fast spin echo) and PROPELLER (periodically rotated overlapping parallel lines with enhanced reconstruction), have been tested, during the last decade, for detection and identification of different pulmonary changes induced by tuberculosis [9,10,11]. These methods are not particularly effective for imaging of lung tissue, characterized by fast signal decay due to very short T_2_/T_2_* relaxation times. In turn, the UTE MRI pulse sequence overcomes this issue since the signal can be acquired starting at tens of microseconds. Therefore, UTE has been successfully applied for ex vivo [15,16] and in vivo [17,18] studies on lung aerosol deposition, and on a study on sponge phantoms simulating lung structure [56].

The main goal of in vitro imaging experiment was to assess MRI contrast possibilities for extremely low MOF concentrations (i.e., to estimate sensitivity) in a highly inhomogeneous environment mimicking lung tissue. The smallest therapeutic INH dose for inhalation was set according to Zhou et al. (2005) as 1% of an oral dose [57]. With this assumption MOF concentration per lung volume was 37.5 µg/cm^3^ (assuming rat weight of 300 g, lung volume of 3 cm^3^, and the fact that MOF constitutes 30% of optimized INH-MOF-loaded PLGA/LC MP blend). This minimal dose was marked in the right panel of Figure 7d. It should be noted, that even for doses below 1% of the oral INH dose the MRI contrast was achieved in terms of image histogram parameters—In this case maximum histogram value. Lowering the maximum histogram value moved more voxel counts towards higher image intensities; i.e., gave positive contrast. However, for the MOF concentration in this range there was no unambiguous dependence of the maximum histogram value on MOF concentration and its values were in the range 0.7–0.9 of that obtained for pure water. But, 10% of an oral dose corresponded to a MOF concentration of 375 µg/cm^3^, which was equal to the highest tested MOF concentration. In this case the maximum histogram value was 0.6 of that obtained for pure water. The 50% of the oral dose was located outside the concentrations range covering the MOF concentration axis in the right panel of Figure 7d. Thus, it means that the most probable inhalable therapeutic INH concentration, which corresponds to 50% of the INH oral dose, would assure excellent contrast [58,59]. A UTE3D imaging pulse sequence with very short TE of 110 micros and short TR of 2345 ms will generate T_1_ weighted images. The effect will be visible even for small or moderate change in T_1_, while even large change in T_2_/T_2_* will not affect MR image much due to the ultra short TE. In our study, the 3D UTE MRI allowed for effective T_1_ weighing for an environment characterized by large spatial magnetic susceptibility gradients; i.e., large air voids and thin water layers inside the sponge—the structure resembling lung tissue.

Contrast possibilities of the optimized INH-MOF-loaded PLGA/LC MP blend (final INH-MOF-loaded PLGA/LC formulation) were consequently confirmed on extracted rat respiratory tract. On one hand, histograms of the MR images of rat lungs followed similar pattern/dependence of MOF concentration as obtained on sponge fantom. On the other hand particles of an optimized blend were able to reach tertiary bronchi structure, and consequently providedthe possibility to visualize them (see the lower panel of Figure 8c). Accounting for, very likely, a not efficient administration method, post mortem administration and the much smaller rat lung structure (as compared with human) indirectly corresponds to impinger results obtained for the optimized INH-MOF-loaded PLGA/LC MP blend. Concerning the size difference, the linear diameter of left (main) bronchia is 9.5 times larger in humans than in rats [60]. For this reason, it should be noted that adopting the inhalatory dry powder formulation to the small animal species should not be recommended. The reason is that aerodynamic properties do not change according to the species, but what is changing is the anatomy, size, and physiological features of the lungs that cannot be completely accounted for when engineering respirable particles. Moreover, the target is always humans; therefore, changing the formulation to adjust it according to animal species would be pointless.

Both MRI experiments, i.e., on sponge phantom and consequent ex vivo experiment, constituted the final step of the complete proof of concept study. The INH concentration in the MRI study, was based on published small animals studies, although the effectiveness of inhalable approach to INH-containing MPs administration has been proven in studies on large animals in vivo. Verma et al. [61] have administered inhaled MPs containing INH to rhesus macaques in the following single dosages: 0.25, 2.5, and 25 mg. They observed that the intracellular INH concentration in alveolar macrophages was below limits for the lowest dose. Since the level of the drug in tissue homogenate is proportional to the level of the drug in macrophages, then, even for the lowest dose tested, the drug will be present in macrophages. Thus, it has been concluded that dose levels between 0.25 and 2.5 mg would be appropriate for humans, which corresponds to the 4% *w*/*w* INH content in the optimized INH-MOF-loaded PLGA/LC MP blend. Verma et al. indicated also that the inhaled dose must be kept at a low level to target alveolar macrophages and especially that dry powder inhalation may cause significant systemic bioavailability. It is noteworthy that there is no guideline to determine the correct dose of INH for inhalation for humans and all attempts to date to determine it relate to a fixed oral dose. Although, taking into account the complexity of inhaled administration and the dependence on patient physiology and pathological changes, determining one correct dose is the wrong approach, as opposed to personalization, which is possible thanks to theranostics. In spite of all, recent studies conducted on monkeys [52] indicate the high potential of inhalation therapy in both local and systemic therapy, or inhalation therapy supporting an oral therapy, each time allowing for dose reduction and increased treatment efficacy. Therefore, our approach may be suitable to meet clinical needs in inhaled TB therapy.

At the end of the discussion it should be noted that further modifications of the obtained, MOF based dosage form to inhalation are possible. Considering that TB treatment requires multi-drug therapy, we want to point out that the solution containing only INH was thought of as a starting platform in order to understand feasibility of INH-MOF theranostic inhalatory formulation. Moreover, the proposed multiparticle approach based on blending has the advantage of allowing the addition of different drugs embedded in different particle formulations by simple mixing of the different delivery systems. By this principle it would be easy to add multiple drugs encapsulated in different particles in order to obtain a multidrug delivery system in which the single drug can be reciprocally combined according to precise ratios.

## 5. Conclusions

The study presents a complete proof of concept of the development of an MOF-based, inhalable theranostic system for tuberculosis treatment. Using INH-MOF nanoparticles as a starting material assuring modified INH release, spray-drying technology allowed us to obtain a theranostic, optimized, INH-MOF-loaded PLGA/LC MP blend with:Good aerodynamic properties combined with controlled release of INH;Improved absorption of the formulation by macrophages;Promising contrast properties enabling monitoring of the distribution of formulation within the inhomogeneous structure of the lungs.

To date, this study represents the first attempt to fabricate MOF-based, inhaled formulation to be administered by a dry powder inhaler, combining drug delivery and magnetic resonance imaging contrast agent functionality.

The successful development of such MOF inhalable theranostic platforms will enable personalized delivery of active substances to the lungs and greatly contribute to the increase of inhaled therapeutic efficacy. In fact, as pulmonary antibiotic treatment requires careful control of the delivered dose, MRI monitoring upon administration may grant important benefits for the development of inhaled TB personalized medicines.

## Figures and Tables

**Figure 1 pharmaceutics-11-00687-f001:**
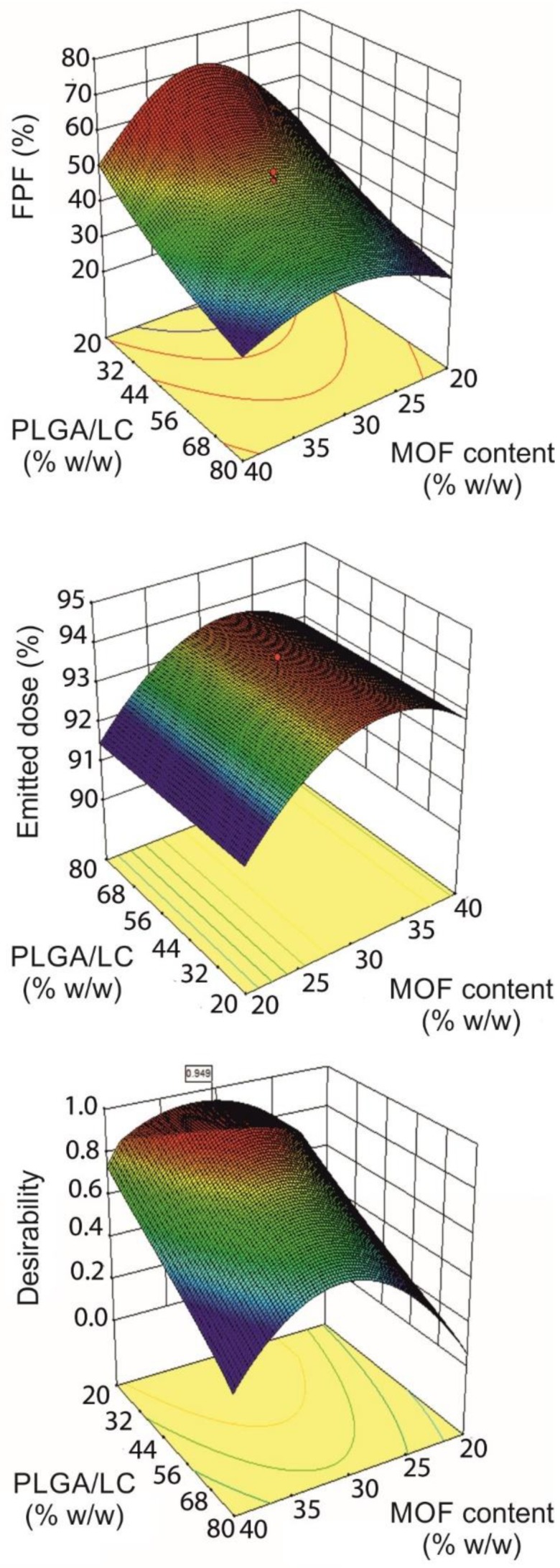
Surface plots depicting the effect of MOF content and the PLGA/LC ratio in %FPF and %ED. The desirability surface plot is also reported, indicating the conditions simultaneously satisfying the requirements of high %FPF and %ED, which grant the desired aerodynamic behavior. Color map: red = high values, blue = low values.

**Figure 2 pharmaceutics-11-00687-f002:**
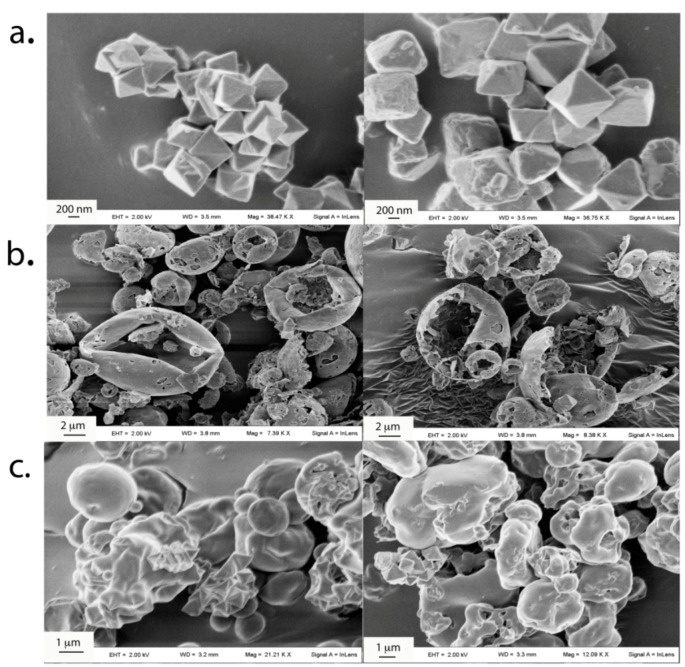
(**a**) SEM images of INH-MOF, (**b**) INH-MOF-loaded LC MPs, (**c**) INH-MOF-loaded PLGA MPs.

**Figure 3 pharmaceutics-11-00687-f003:**
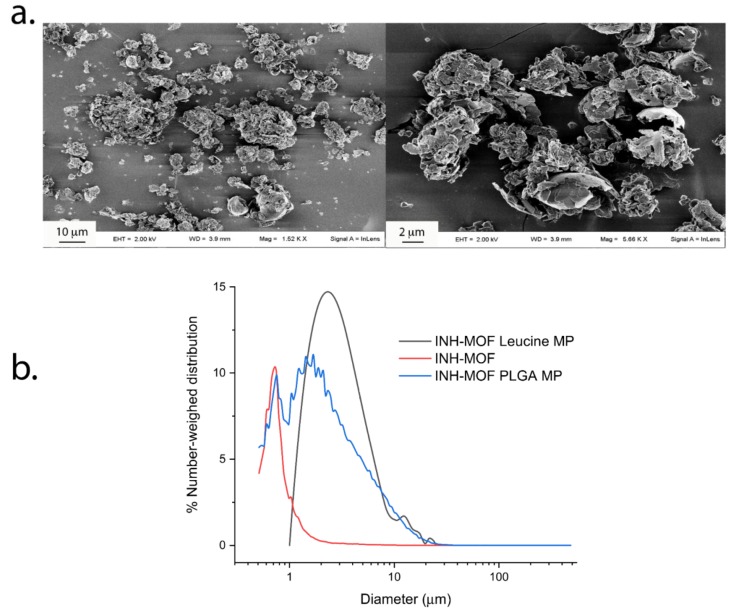
(**a**) Particle size distributions of INH-MOF, INH-MOF-loaded LC MPs, and PLGA MPs; (**b**) SEM images of the optimized INH-MOF-loaded PLGA/LC MP blend.

**Figure 4 pharmaceutics-11-00687-f004:**
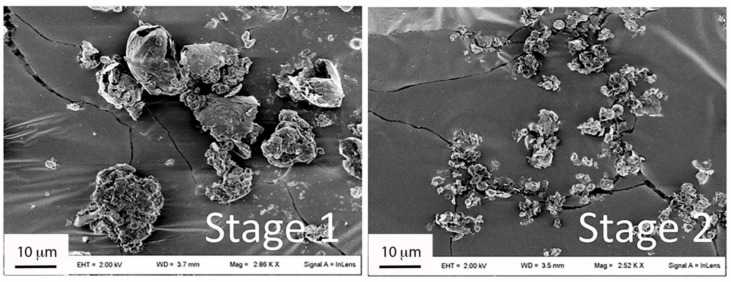
SEM images of the dispersed optimized INH-MOF-loaded optimized PLGA/LC MP blend—Powders deposited in the stage 1 and stage 2 of a twin stage impinger (TSI). The 2nd stage of TSI corresponds to the lower lung part in which the drug is intended to act.

**Figure 5 pharmaceutics-11-00687-f005:**
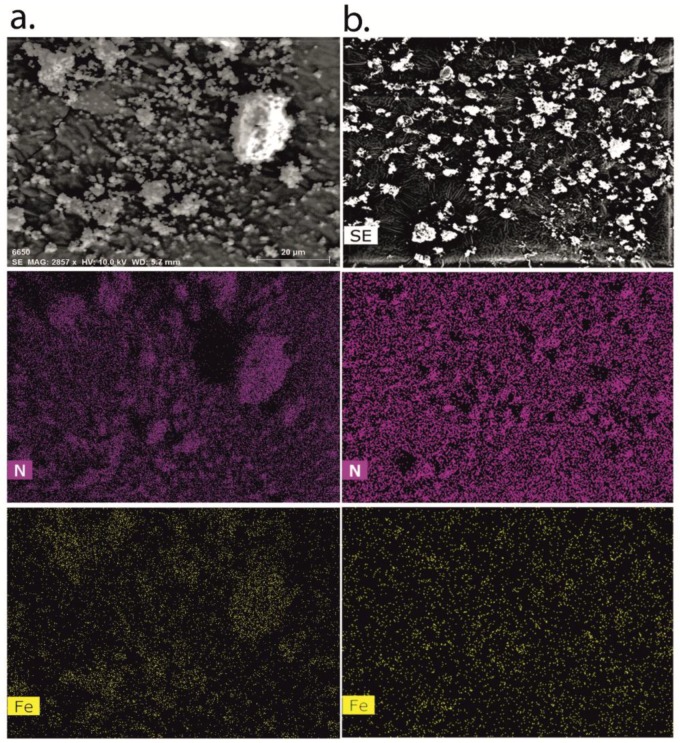
(**a**) EDX analysis of INH-MOF; (**b**) optimized INH-MOF-loaded PLGA/LC MP blend. Mapping shows nitrogen (purple) and iron (yellow) distributions.

**Figure 6 pharmaceutics-11-00687-f006:**
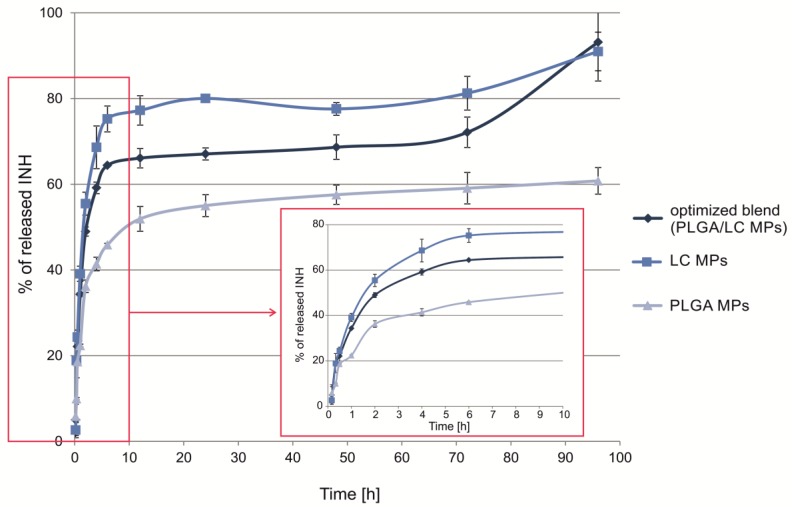
In vitro INH release profiles from spray-dried INH-MOF-loaded PLGA MPs, LC MPs, and the optimized, INH-MOF-loaded PLGA/LC MP blend (0.1 M phosphate buffer, pH 7.4, 37 °C; *n* = 3). Inclusion: magnification of the initial part of the dissolution profile.

**Figure 7 pharmaceutics-11-00687-f007:**
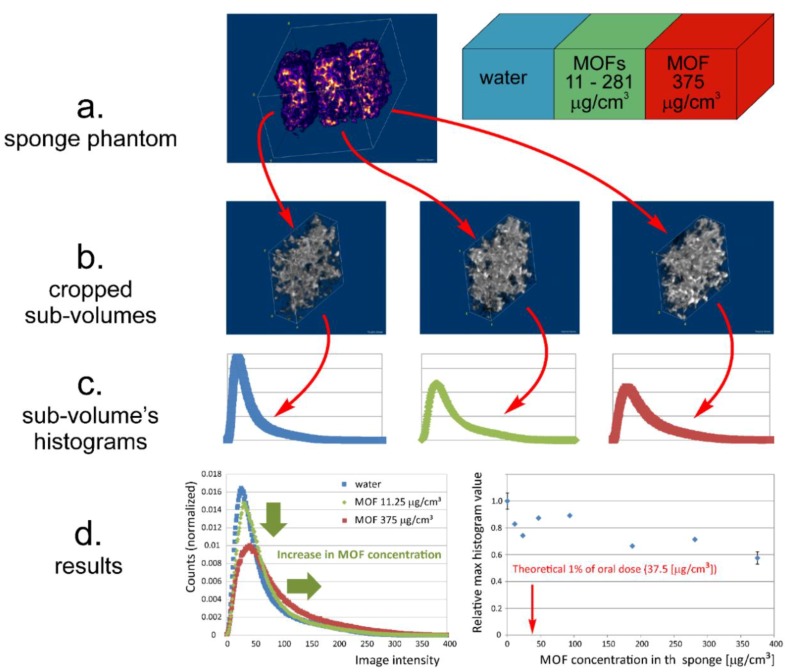
Qualitative and quantitative analysis of MR volume images of sponge phantom. (**a**) Sponge phantom diagram and fragment of the MRI image volume; (**b**) cropped sub-volumes; (**c**) histograms of sub-volumes of the sponge phantom; (**d**) example of the resulted histograms obtained using exchangeable sponge phantom wetted with a 0.03 mg/mL MOF suspension (11.25 µg of MOF per cm^3^ of sponge volume)—Green arrows show the influence of MOF addition on the image intensity histogram; i.e., decrease in maximum histogram value and expansion in the image intensity range towards higher intensities (left); histogram maximum value versus MOF concentration in the sponge—The red arrow points at the smallest therapeutic INH dose for inhalation per lung volume of 37.5 µg/cm^3^, assuming rat weight of 300 g, lung volume of 3 cm^3^ and the fact that MOF constitutes 30% of optimized INH-MOF-loaded PLGA/LC MP blend (right).

**Figure 8 pharmaceutics-11-00687-f008:**
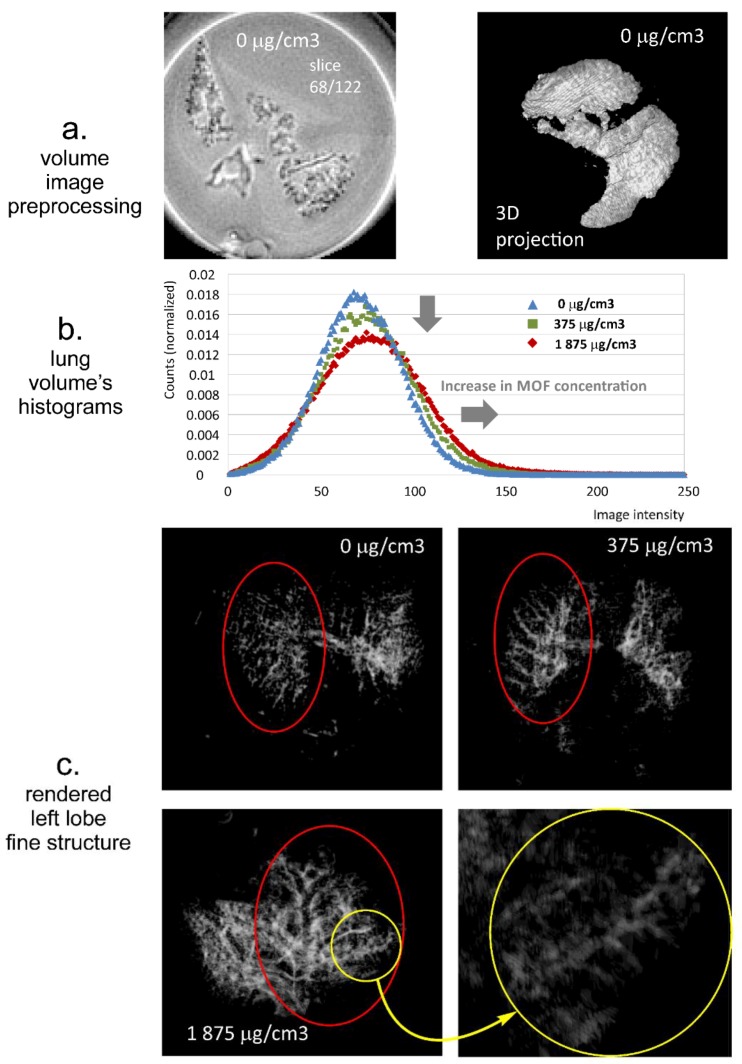
Qualitative and quantitative analysis of MRI volume images of lungs. (**a**) Single slice from the MRI image volume (left) and 3D projection of segmented lung image volume (right); (**b**) histograms of volume lung images obtained for three different MOF concentrations (the grey arrows show the influence of MOF concentration on the image intensity histogram; i.e., decrease in maximum histogram value and expansion in the image intensity range towards higher intensities); (**c**) views of rendered left lobe fine structure (the yellow circle shows details of deeper lung areas contrasted with MOF).

**Figure 9 pharmaceutics-11-00687-f009:**
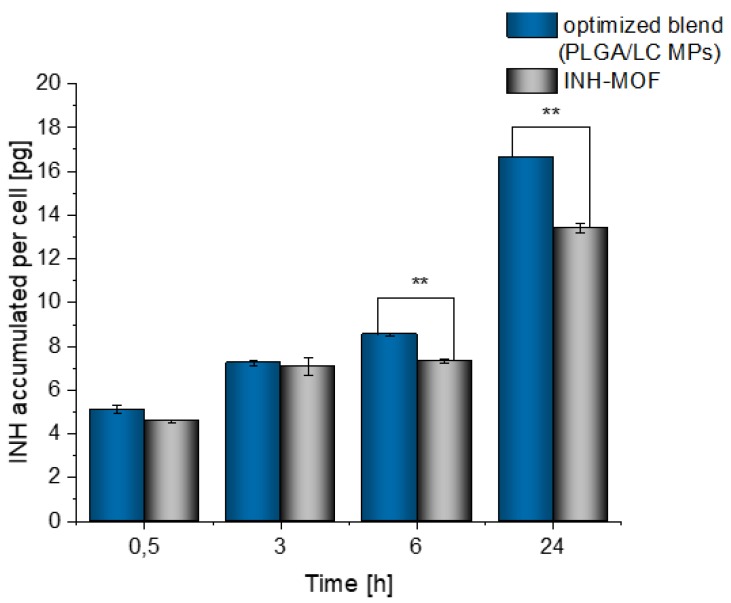
Comparison of the INH amount [pg] accumulated in macrophages treated by optimized INH-MOF-loaded PLGA/LC MP blend and INH-MOF. The amount of INH per cell was calculated based on the relative number of cells designated using the ToxiLight test. The mean values and error bars are defined as the mean and SD, respectively. ** *p* < 0.01, as determined by unpaired two-tailed Student’s *t*-tests.

**Figure 10 pharmaceutics-11-00687-f010:**
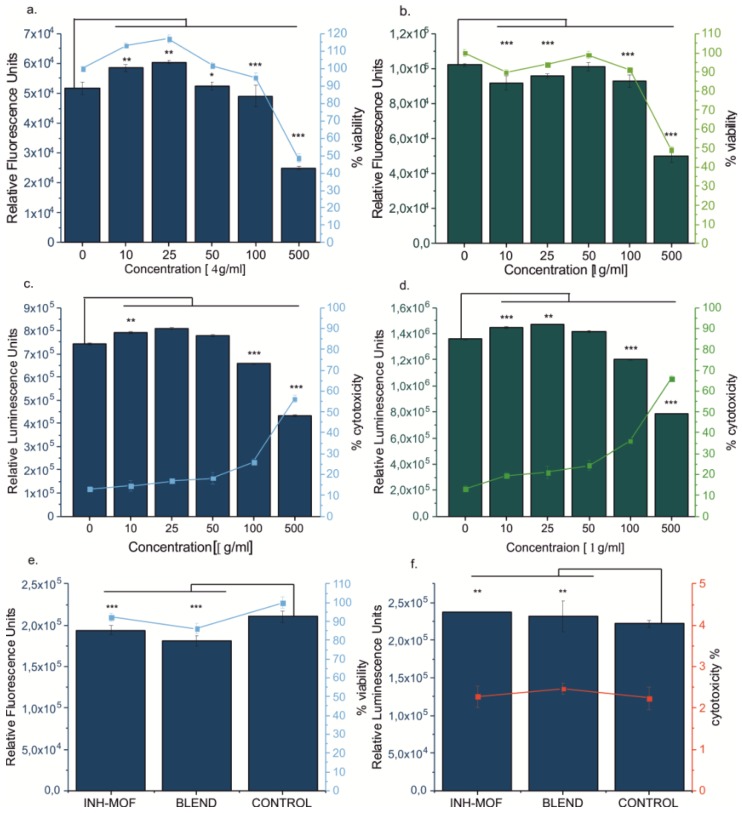
Macrophages viability determined by PresoBlue^TM^ test: dependence of fluorescence (in relative fluorescence units) on the concentration of MOF after (**a**) 24 and (**b**) 72 h. Cytotoxicity of the MOF on macrophages cells after incubations of (**c**) 24 and (**d**) 72 h. The mean values and error bars are defined as mean and SD, respectively. * *p* < 0.05, ** *p* < 0.01, *** *p* < 0.001, as determined by unpaired, two-tailed Student’s *t*-tests. (**e**) Viability of macrophages treated by INH-MOF and the optimized INH-MOF-loaded PLGA/LC MP blend; (**f**) Cytotoxicity effect of INH-MOF and the blend evaluated based on the luminescence intensity of supernatant and lysate. The mean values and error bars are defined as the mean and SD, respectively. ** *p* < 0.01, *** *p* < 0.001, as determined by unpaired two-tailed Student’s *t*-tests.

**Figure 11 pharmaceutics-11-00687-f011:**
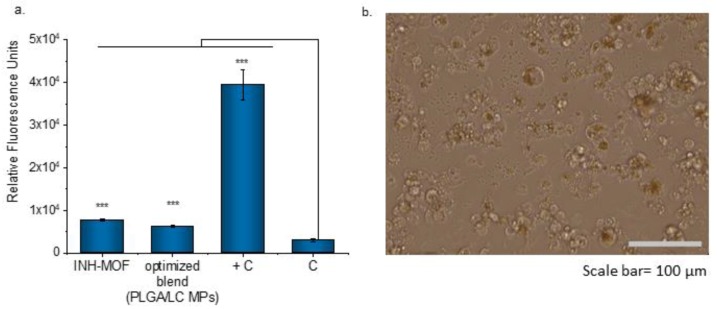
(**a**) Average ROS/radical concentrations for H-MOF and the blend in comparison with positive control (+C) (100 μM solution of H_2_O_2_ treated) and untreated cells (C). The mean values and error bars are defined as the mean and SD, respectively. *** *p* < 0.001 as determined by unpaired, two-tailed Student’s *t*-tests; (**b**) morphology of macrophages contacted with the blend for 24 h.

**Table 1 pharmaceutics-11-00687-t001:** 2^3^ factorial design for the development of spray-dried INH-metal-organic framework (MOF)-loaded microparticle (MP) blend and the respective response parameters. For the preparation a hydrophobic and hydrophilic excipient were used: PLGA and LC. The blends were obtained changing MOF amount, the PLGA/LC ratio, and blending time. The aerodynamic behavior of the blends was assessed through the TSI test by measuring fine particle fraction (%FPF) and emitted dose (%ED). The content homogeneity of the blends corresponding to each run is also reported.

Run	Factors				Responses		Content Homogeneity *p*-Value (*n* = 9)
	A	B		C			
	INH-MOF Content [% *w*/*w*]	Blending Ratio [% *w*/*w*]		Blending Time [min]	%FPF	%ED	
		PLGA	LC				
1	20	20	80	5	48.06	91.29	0.078
2	20	80	20	5	25.99	90.91	0.840
3	20	20	80	15	46.50	91.90	0.711
4	20	80	20	15	26.86	91.85	0.368
5	40	20	80	5	47.17	93.19	0.103
6	40	80	20	5	28.18	92.00	0.483
7	40	20	80	15	54.78	93.30	0.069
8	40	80	20	15	28.06	92.97	0.380
9	30	50	50	10	52.29	94.01	0.740
10	30	50	50	10	45.75	93.35	0.192
11	30	50	50	10	49.43	93.50	0.128

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
