# Peer review of "An Inhalable Theranostic System for Local Tuberculosis Treatment Containing an Isoniazid Loaded Metal Organic Framework Fe-MIL-101-NH2—From Raw MOF to Drug Delivery System"

_pharmaceutics, 2019, doi:10.3390/pharmaceutics11120687_

Round 1

Reviewer 1 Report

This manuscript addresses an actual theme, which is the development of innovative therapeutics to fight TB. A proof of concept of a hybrid system capable of targeting macrophages, and possessing the ability of acting as contrast  in MRI as well as delivering an anti-tubercular drug (isoniazid), is presented. The manuscript is well structured and well written, however there are some aspects that need to be addressed before consideration for publication.

1. Starting with the title, the term “nanoparticles” is not quite adequate, since the size of MOF is near 1 um; it also should read “… to a drug delivery system”.

2. The authors used only one drug (isoniazid) in the development of their system. However, TB treatment is performed by combination therapy. Therefore, it seems a little pointless developing a single-drug delivery system for a pathology that is treated by a multidrug combination.

3. The authors state (line 106) that they will prepare and blend hydrophilic and hydrophobic MPs loaded with INH-MOF. However, they don’t present any rationale for this.

4. DMF was used as solvent in the incorporation of INH in MOF. The loaded particles were separated from the supernatant by centrifugation and no drying was performed. Being DMF a high boiling point solvent, that presents many hazards (respiratory sensitization and carcinogenicity, among others), the continuous exposure of patients to its residues throughout treatment would be detrimental. This effect would not be detected in the performed cytotoxicity evaluation (exposure of cells to one single dose for a 24 h period).

5. The second sentence in 2.4 (lines 133-135) should switch with the first (lines 131-133).

6. Equations in lines 158 and 159 should be numbered (3) and (4), following the numbering of equations in lines 138 and 144.

7. The point “MOF and drug quantification” lacks information: in which solvent was MOF dissolved for quantification? Was INH entrapped in MOF quantified?

8. For the drug release profile (described starting in line 186) the authors used a dialysis method. However, this method presents a drawback: the drug release into the outside medium is controlled by permeation across the dialysis membrane. Therefore, the slow appearance of the drug in this medium may be mistaken with a controlled release.

9. Line 214: a sentence never starts with a number. It should be re-written or the number spelled-out.

10. In lines 318-319 the authors say that “The encapsulation of MOF crystals produced fragile MPs, in particular LC MPs, that were partially fragmented even after blending”. Of course, if particles were fragmented before, they remain fragmented after blending.

11. In lines 321-322 the MPs diameters should be in growing order. Moreover, from figure 2.2 it can be seen that in INH-MOF PLGA MPs there are two populations: one corresponding to the particles and another corresponding to free MOF, either to lack of polymer to cover all the MOF or MOF released from the fragmented particles. This observation should be included in the text here and in the discussion. Also, data in graph 2.2 should include the blend, as the mixture of both MPs can lead to the formation of aggregates and to a size distribution different from the sum of those of individual MPs.

12. SEM images presented in figure 2 are too small. The image details are hardly noticed in the micrographs of figure 3.3 and the scale bars are not visible in none of the micrographs. A size similar to that of micrographs in figure 3 should be selected.

13. In lines 339-340 it is said that “As reported in Table 1, a good content uniformity was measured either in the blends or the PLGA and LC MPs (%RSD was in the range of 2-7)…”. However, no such information is presented for PLGA- and LC-only MPs.

14. In the legend of the graph on the left panel of figure 6.4, the values should come in increasing order. Regarding the graph on the right panel of the figure, the trend line makes no sense and should be removed. In fact, although there is a decreasing trend for higher concentrations, no such trend is observed for lower ones, as pointed by the authors later on (lines 550-552). Moreover, point (0,1) is an experimental point (relative histogram max for water) and should be included. This point and that corresponding to the highest MOF concentration, which were performed 6 times, should bear error bars. Perhaps it would make more sense to represent the image intensity at the histogram maximum as a function of MOF concentration, instead of representing the maximum value of the histogram. Also, this graph should be extended to the maximum concentration of MOF (1875 ug/cm3) in order to support the discussion starting in line 549. The theoretical value for 1% of oral dose (75 ug/cm3) is not in line with the text (line 544), which indicates this value as being 37.5 ug/cm3. The caption of figure 6 needs revision: the explanation regarding the green arrows (last sentence) refers to the left panel on figure 6.4 and therefore should be moved accordingly; in lines 372 and 374 the term “results -” should be removed; information regarding the meaning of the red arrow and corresponding text in the right panel should be added.

15. The sentence of 390-391 makes no sense.

16. The caption of figure 8 should refer which data are presented by the bars and which are presented by the line.

17. Line 493 should read: “Results (Figure 8)…”

18. The presented data do not prove that both types of MPs present in the blend are captured by macrophages, as both are loaded with the same drug. In fact, regarding their size (6.7 um), way above the size that potentiates macrophage capture (1 to 3 um), and water solubility, it is very likely that LC MPs do not enter those cells, rather dissolving in the culture medium and releasing MOF. This does neither get captured, as proved by the authors in their previous work, hence the need for the coverage developed in the present study. In order to prove the capture of both MPs present in the blend, a test comprising a blend where each type of MP is loaded with a different compound, say two fluorescent dyes, and the presence of both dyes inside the cells qualitatively verified or quantified by an adequate method, such as fluorimetry, confocal microscopy or flux cytometry, for instance, should be performed.

19. Line 546 should read: “… in the right panel…”

20. Line 557: Don’t the authors mean T2 weighing?

21. Line 561 should read: “On one hand…”

22. The discussion held in lines 546-559 is quite confusing and should be re-written for clarification of what is meant by the authors.

Reviewer 2 Report

This is an interesting manuscript and I think the authors may like to consider the following points.

Lines 111-116.  This is all in one sentence this needs to be split.

Line 132           23factorial should be 23 factorial

Line 194-199     Did the authors check that the data collected met the criteria for use of parametric statistics?  I am unclear from the description of the statistical methods how treatments analysed using an ANOVA were then compared using a post-hoc test.  It is incorrect to carry out multiple comparisons for treatments within an experiment using t tests.  A t test should only be used to compare data in an experiments that has two treatments.

I do not see an ethics statement for the use of rats in this study.  I imagine that rats were used as part

of a schedule 1 procedure, but I think that an ethics statement on information on how rats were house

is required. What age and sex of rats were used?

Line 251           What is 200.103 cells/well?

Line 340           Data is significantly different or not – please remove the statement that ‘with nearly statistically significant sampling differences (p < 0.078 and p < 0.069)’ and rewrite this section.

Fig. 5                Are there any significant differences between treatments at the different time points? These should be shown the Fig. What is the n value/treatment used? The n value should be shown in the other data too.

Fig. 6                A linear line has been fitted to the data shown on the right but it is obvious from the correlation coefficient used that the data does not fit this model. Did the authors try any other type of curve fitting? If a relationship is going to be discussed then the correct model must be applied.

Fig. 8                the legend must be on the same page as the Fig. 8 – 2b is missing the significant difference shown for the 500 µg/ml treatment. There are more than two treatments in this experiment so a t test is not appropriate for analysing this data. I think the authors need to redo the statistics as some treatments do not look very different e.g. Fig.8-3b

Line 452           I think the phrase ‘no negative influence on cell viability was observed’ should be rewritten. A treatment either caused a significant reduction in cell viability or had no effect on cell viability.

Line 468                   Macrophages are end cells and do not proliferate.  The RAW cells should not be proliferating in experiments where they are being used as a macrophage model.

Line 467                   The statement ‘slightly decreased’ should be deleted – you either have no significant effect or a significant effect.

Round 2

Reviewer 1 Report

The authors have correctly addressed the issues pointed by the reviewer. Therefore, in its presente format, the manuscript is ready to be considered for publication.